# A High-Throughput Screening System Based on Droplet Microfluidics for Glucose Oxidase Gene Libraries

**DOI:** 10.3390/molecules25102418

**Published:** 2020-05-22

**Authors:** Radivoje Prodanović, W. Lloyd Ung, Karla Ilić Đurđić, Rainer Fischer, David A. Weitz, Raluca Ostafe

**Affiliations:** 1Faculty of Chemistry, University of Belgrade, Studentski trg 12, 11000 Belgrade, Serbia; karlailic@chem.bg.ac.rs; 2Department of Physics, School of Engineering and Applied Sciences, Harvard University, Cambridge, MA 02138, USA; lloyd.ung@gmail.com (W.L.U.); weitz@seas.harvard.edu (D.A.W.); 3Departments of Biological Sciences and Chemistry, Purdue University, 207 S. Martin Jischke Dr., West Lafayette, IN 47907, USA; fische70@purdue.edu; 4Purdue Institute of Inflammation, Immunology and Infectious Disease, Molecular Evolution, Protein Engineering and Production, Purdue University, 207 S. Martin Jischke Dr., West Lafayette, IN 47907, USA; rostafe@purdue.edu

**Keywords:** fluorescent label, sorting, protein engineering, enzyme optimization

## Abstract

Glucose oxidase (GOx) is an important industrial enzyme that can be optimized for specific applications by mutagenesis and activity-based screening. To increase the efficiency of this approach, we have developed a new ultrahigh-throughput screening platform based on a microfluidic lab-on-chip device that allows the sorting of GOx mutants from a saturation mutagenesis library expressed on the surface of yeast cells. GOx activity was measured by monitoring the fluorescence of water microdroplets dispersed in perfluorinated oil. The signal was generated via a series of coupled enzyme reactions leading to the formation of fluorescein. Using this new method, we were able to enrich the yeast cell population by more than 35-fold for GOx mutants with higher than wild-type activity after two rounds of sorting, almost double the efficiency of our previously described flow cytometry platform. We identified and characterized novel GOx mutants, the most promising of which (M6) contained a combination of six point mutations that increased the catalytic constant k_cat_ by 2.1-fold compared to wild-type GOx and by 1.4-fold compared to a parental GOx variant. The new microfluidic platform for GOx was therefore more sensitive than flow cytometry and supports comprehensive screens of gene libraries containing multiple mutations per gene.

## 1. Introduction

Enzymes are widely used for industrial processes, but natural enzymes tend to be incompatible with harsh process conditions and must be optimized using molecular evolution methods to improve properties such as stability or substrate preference [1,2]. This protein engineering approach involves the generation of gene libraries containing variants of the original enzyme sequence, followed by screening to select enzyme variants with desirable characteristics [1]. The greater the diversity in the library, the more likely a better-performing mutant will be identified. However, the link between the mutant phenotype and genotype must be preserved during screening in order to recover the corresponding gene sequence for enzyme expression [3]. The link is easily preserved by screening in microtiter plates if each variant is restricted to a single well, but this approach limits the library complexity to a maximum of 10^4^ variants per screening round, increasing the likelihood that the most active or stable enzyme variants will be overlooked [4]. Higher-throughput methods are therefore needed to screen libraries containing 10^6^–10^10^ variants, which is the scale required to capture enzyme structural diversity in its entirety.

Fluorescence-activated cell sorting (FACS) has been used with success to screen horseradish peroxidase (HRP) mutants for modified enantioselectivity [5]. The enzyme was expressed on the surface of yeast cells, where it converted a covalently-bound substrate into a fluorescent product. This approach becomes challenging with more complex reactions due to the potential for cross-labeling between cells expressing different enzyme variants [6]. Most FACS screening protocols therefore involve reactions performed in double (water-oil-water) emulsions [7], which provide individual compartments to preserve the link between enzyme activity and the corresponding gene. However, the size of the emulsion droplets is difficult to control, leading to differences in fluorescence levels due to the presence of more substrate in the larger droplets, and false positives may occur if multiple inner droplets with different enzyme variants are encapsulated together [8]. FACS sorting for the directed evolution of enzymes could be improved greatly if there were a simple way to produce monodisperse double emulsions.

Microfluidics offers a potential solution because this state-of-the-art technology significantly reduces fluid volumes, allowing the production of monodisperse emulsion droplets whose size and composition can be controlled [8]. Microfluidic lab-on-chip (LOC) devices are attractive due to their low cost, fast reaction times, and high sensitivity [9]. Such devices are therefore found in diverse applications such as next-generation sequencing [10], environmental studies [11], drug discovery, organ-on-a-chip system development [12], and cancer diagnostics [13]. Microfluidics is particularly suitable for cell-based assays [14] and single-cell analysis [15], facilitating the development of high-throughput screening systems [16].

Another challenge associated with enzyme screening is that high enzyme concentrations within an emulsion droplet can rapidly exhaust the substrate, taking the assay to plateau in a matter of seconds or minutes. Once the assay has plateaued, any correlation between enzyme activity and fluorescence intensity is lost and there will be no measurable difference between droplets containing enzyme variants with high and low activity. This problem is exacerbated by FACS because the timeframe between emulsion preparation and screening/sorting is difficult to control [17]. The use of competitive enzyme reactions with two substrates can partially address this problem, as demonstrated in a screen for HRP variants [18], but this solution is not applicable for all assays. However, the problem is minimized by microfluidic LOC technology because the timing of cell encapsulation, mixing with substrate, signal measurement, and sorting can be precisely controlled. This allows an enzymatic reaction to be characterized after time delays of milliseconds, seconds or minutes [8,19]. Furthermore, the smaller reaction volumes and use of disposable microchips significantly reduce the cost of the process [9].

Given these combined advantages, the high-throughput screening of enzyme libraries by droplet microfluidics should allow the top 5% most active enzyme variants to be sorted from a library containing only active mutants. We applied this approach to optimize the properties of glucose oxidase (GOx), a flavoprotein monooxygenase (β-d-glucose: oxygen 1-oxidoreductase, EC 1.1.3.4) that catalyzes the oxidation of β-d-glucose to glucono-δ-lactone, which in turn spontaneously hydrolyses to form gluconic acid. GOx is naturally secreted by *Aspergillus niger* as a glycosylated homodimer with a molecular weight of 160 kDa [20]. The enzyme has many commercial applications [21] in the food [22,23,24], pharmaceutical [21,25], and textile [26] industries, in biofuel production [27] and, perhaps most importantly, in the construction of biosensors to measure blood glucose concentrations [28]. GOx variants with optimized kinetic parameters are valuable because they may increase the sensitivity of glucose detection while boosting output power [25]. Several previous reports therefore describe protein engineering and high-throughput screening methods to increase the activity [7,17,25] and stability [6,29] of GOx.

During the oxidation reaction, GOx reduces molecular oxygen to hydrogen peroxide, which can in turn be used to measure GOx activity [30]. Accordingly, we adapted a previously reported vanadium bromoperoxidase-coupled fluorescence assay (ViPer) for the detection of GOx activity and developed an ultrahigh-throughput screening system based on droplet microfluidics to sort GOx gene libraries and find variants with improved activity. Using this new system, we discovered a novel GOx mutant with six point mutations conferring a 2.1-fold increase in the catalytic constant k_cat_ compared to wild-type GOx.

## 2. Results

The novel droplet microfluidics assay for GOx activity takes hydrogen peroxide produced by GOx during the oxidation of glucose and uses it as a substrate for vanadium bromoperoxidase, which combines hydrogen peroxide with bromide to produce hypobromite radicals. These radicals are detected when they react with aminophenylfluorescein to produce fluorescein (Figure 1). The accumulation of fluorescein in droplets containing a single yeast cell expressing a particular GOx variant is easily detected by the optical system of the droplet sorter, with higher fluorescence intensity correlated with higher GOx activity. Thresholds can be applied to sort droplets with high fluorescence in real time, such that the cells with high-performing GOx variants are retained for further analysis.

The use of vanadium bromoperoxidase rather than the more widely used HRP to couple the detection of hydrogen peroxide to fluorescein production prevents a drop in fluorescence due to the oxidation of fluorescent coumarin products by HRP in the presence of excess peroxide. Vanadium bromoperoxidase does not contain a heme group and thus cannot oxidize phenolic substrates, which ensures that vanadium bromoperoxidase assays achieve strong correlation between enzyme activity and fluorescence [17].

An initial round of reference library sorting (using defined mixtures of cells expressing or not expressing GOx) was required to determine the optimal laser position to ensure that the correct droplet was captured by the sorting system each time a sorting burst was triggered. When the optimal laser position was identified, a site-directed mutagenesis library of GOx sequences with mutations at 16 positions [6] (Appendix A) was sorted several times to isolate droplets with the most intense green fluorescence.

The library was created using consensus approach by multiple-site-directed mutagenesis at all 16 positions simultaneously (Figure 2) and contained 10^5^ different GOx mutants among ~10^7^ sorted transformants. Recordings of the GOx library before and after sorting are shown in Figure 3.

The retained variants were transferred to agar plates immediately after sorting, and colonies derived from each sorted cell were tested for GOx activity using a standard 2,2′-azino-bis(3-ethylbenzothiazoline-6-sulfonic acid) (ABTS) assay, in which GOx activity is revealed as a colored halo surrounding the cells expressing an active GOx variant. This provided an initial readout for the enrichment of each sorted library. We also conducted detailed kinetic analysis of the GOx variants in microtiter plates in order to quantify the enzyme activity and classify the variants as inactive, less active than wild-type GOx, or more active than wild-type GOx (Figure 4). After two rounds of sorting, the proportion of cells in the population expressing GOx variants with higher than wild-type activity exceeded 60% which compares favorably to the 36% obtained after sorting the same library by flow cytometry [6]. After the third round of sorting under the same conditions, the proportion of variants with higher than wild-type activity increased to 69%, and we could no longer detect any cells expressing inactive GOx.

The most active enzyme variants in the enriched library were sequenced, recloned for expression in *Pichia pastoris*, purified, and characterized in more detail (Table 1). The most active mutant in the enriched library (A2) and three others (A21, F9 and F91) were also recovered by flow cytometry in our previous study [6].

However, two new mutants (M6 and M15) were isolated for the first time in the current study. M6 was the most active novel variant, and featured point mutations at six positions (Figure 5).

The catalytic constant of M6 was 2.1-fold higher than the wild-type enzyme and 1.4-fold higher than its parent GOx variant (T30V, I94V and A162T).

## 3. Discussion

The sorting of gene libraries in a directed evolution experiment requires a physical connection between the enzyme activity (phenotype) and the corresponding gene sequence (genotype). The reactions must therefore be compartmentalized, and there must be no cross-talk between reaction compartments caused by the diffusion of products. Furthermore, the reaction volume must be reduced as far as possible to maximize the number of variants that can be screened in a short period of time. These combined requirements led to the development of screening platforms based on yeast surface display and in vitro compartmentalization. The enzyme reaction takes place inside water microdroplets dispersed in an oil phase, which enables the rapid sorting of microdroplets by flow cytometry. However, the diffusion of reaction products can lead to false positives, reducing the sorting efficiency [7].

We developed the ViPer assay for GOx gene libraries in order to overcome some of the challenges discussed above. The ViPer assay releases the charged fluorescent product carboxy-coumarin, hence flow cytometry can be used to sort gene libraries in double emulsions prepared using light mineral oil and ABIL EM90 detergent [17]. One drawback of this approach is the heterogeneity of the double emulsion droplets, which means that fluorescence intensity can vary due to differences in compartment size as well as enzyme activity. The screening results must therefore be normalized for different reaction volumes in order to optimize the sorting efficiency, for example by tagging GOx expressed on the surface of yeast cells with green fluorescent protein [31].

Microfluidics can address the issue of heterogeneity at source by generating monodisperse microdroplets with little variance in reaction volume between reaction compartments, allowing the precise measurement of enzyme activity without normalization. We therefore adapted our ViPer assay to make it compatible with a microfluidic system and the associated perfluorinated oils and detergent. The flow of liquids in microchannels is highly sensitive to changes in liquid composition and LOC manufacturing, so we optimized the laser deflection point to increase sorting efficiency. We then tested the same GOx gene library previously sorted by flow cytometry [6] in order to compare the sorting efficiencies of the two systems. After two rounds of sorting, we found that the proportion of enzyme variants with higher than wild-type GOx activity increased from 6% to 36% when using the original flow cytometry method, but from 2% to 61% using the new microfluidic system. In our previous work, the proportion of active enzyme variants in an error-prone PCR GOx gene library after two rounds of sorting by flow cytometry using the same ViPer assay was 28.12% [17], whereas using our new microfluidic system increased the percentage of active enzyme variants to 97%. The new microfluidics system outperformed flow cytometry due to the monodispersity of the emulsion and the precise incubation time of the enzyme reaction. These results also suggest that fluorescein does not diffuse through the fluorocarbon oil phase.

Although microfluidic systems clearly offer advantages for the directed evolution of enzymes, there are few examples discussed in the literature. Fluorescence activated droplet sorting on microfluidic devices has been used for the directed evolution of HRP [19] and esterase [32], but in both cases the aim was to modify the specificity or enantioselectivity of the enzymes by comparing their activity against different fluorogenic substrates. This facilitates the identification of superior variants because neither the microdroplet volume nor the level of enzyme expression influences the ratio of two fluorescent signals in the same compartment. Fluorescence-activated electrocoalescence has been used as part of an in vitro translation system to sort a galactosidase reference library into positive droplets containing active enzymes and negative droplets containing inactive variants [33]. Similarly, we previously used a microfluidic system to sort a cellulase reference library into active and inactive variants [8]. In the current study, the scale of the challenge was increased by using microfluidics to sort the most active variants from a large background of mutants with lower but still measurable levels of enzyme activity.

Several challenges remain to be addressed before microfluidics can be used routinely for the directed evolution of enzymes and protein engineering. Most fluorescence activated droplet sorting systems are highly sensitive to the fluctuation of flow rates at device junctions caused by cell aggregates or other debris, as well as the pulsatile nature of syringe pumps, although these engineering problems can be solved prior to commercialization using current technology. Another drawback is that flow cytometry can sort droplets into multiple channels, whereas microfluidics is restricted to binary sorting. Like flow cytometry, most microfluidic devices rely on the detection of enzymatic activity based on fluorescence, so a fluorogenic assay must be available for the enzyme of interest. However, attempts to combine mass spectrometry-based detection with microfluidics for the directed evolution of transaminase are promising, and this could enable the use of microfluidics for the directed evolution of enzymes with phenotypes that cannot be coupled to fluorescent readouts [34].

In conclusion, we used a new microfluidics platform for GOx sorting to sort a library of cells expressing variants of the enzyme GOx, and isolated those with higher than wild-type activity from a large background of cells expressing active enzyme variants. The isolation of better-performing enzyme variants was much more efficient using the microfluidics platform compared to our previous flow cytometry method. We identified the novel GOx mutant M6, which was not previously isolated by flow cytometry. This mutant contains six point mutations and its catalytic constant k_cat_ is 2.1-fold higher than wild-type GOx and 1.4-fold higher than the parent GOx variant. The microfluidic LOC screening platform for GOx was therefore more sensitive than the flow cytometry system and can be used for more comprehensive screens seeking active variants in GOx gene libraries containing multiple mutations per gene.

## 4. Materials and Methods

### 4.1. Multiple-Site-Directed Mutagenesis

The GOx saturation gene library was prepared using the QuickChange multiple-site-directed mutagenesis kit (Agilent Technologies, Santa Clara, CA, USA). Primers were used to mutate 16 positions in the GOx gene sequence (Appendix A). The PCR mix contained 200 nM of the 16 primer mix and 400 pg/μL of template DNA (GOx in vector pCTCON_2_). The reaction mix was heated to 95 °C for 1 min, followed by 30 cycles of 95 °C for 1 min, 55 °C for 1 min and 65 °C for 16.5 min, and a final extension step at 65 °C for 20 min. The template DNA was then digested with DpnI for 3 h at 37 °C. The resulting single-stranded plasmid DNA was introduced into *E. coli* XL10 Gold ultracompetent cells (Agilent Technologies, Santa Clara, CA, USA) according to the manufacturer’s instructions. The library quality and the introduced mutations were checked by sequencing randomly-selected clones.

### 4.2. Expression of GOx on the Surface of Yeast Cells

*Saccharomyces cerevisiae* EBY 100 cells (kindly provided by Prof. Dane Wittrup, MIT, Boston, MA, USA) were transformed with plasmid DNA as previously described [35] by applying a 42 °C heat shock for 2.5 h. The cells were transferred to YNB-CAA Glu medium for recovery, and were incubated at 27 °C, shaking at 160 rpm for 48 h. GOx expression was induced by transferring the cells to YNB-CAA Gal/Raf medium and incubating as above for 16–18 h. Following induction, the cells were washed three times with 50 mM Tris (pH 7.4) containing 0.01% Triton X-100 and were stained with mouse anti-cMyc primary antibodies and with goat anti-mouse IgG DyLight 550 secondary antibodies as previously described [6].

### 4.3. Fabrication of Microfluidic Devices

The polydimethylsiloxane microfluidic devices were prepared by soft lithography using SU8-on-Si wafer masters and PDMS-on-glass devices as previously described [19]. Electrodes were made by filling channels in the PDMS device with a low-melting-point alloy, by pushing the wire of Indalloy 19 (52 In, 32.5 Bi and 16.5 Sn) into punched holes. Electrical connections were made using eight-pin terminal blocks (DigiKey, Thief River Falls, MN, USA) glued (Loctite, UV cured, Henkel, Disseldorf, Germany) to the surface of the device.

### 4.4. Emulsification and Sorting

Monodisperse aqueous drops were prepared in fluorocarbon oil using a flow-focusing geometry with three streams. The first stream was Novec HFE 7500 fluorocarbon oil (3M, St. Paul, MN, USA) containing 1% (*v/v*) Krytox-PEG-Krytox detergent synthesized as previously described [36]. The second stream was an aqueous solution containing a defined mixture of library cells (*S. cerevisiae* EBY100 cells) transformed with wild-type GOx (positive cells) or the empty pCTCON2 vector (negative cells). The third stream was the aqueous substrate solution, consisting of 100 μM aminophenoxyfluorescein (Invitrogen, Carlsbad, CA, USA) 200 mU/mL *Corallina officinalis* VBrPOx (Sigma-Aldrich, Taufkirchen, Germany), 200 mM glucose, 10 mM NaBr and 1.5 mg/mL bovine serum albumin in 50 mM Tris-H_2_SO_4_ (pH 7.4).

Two aqueous streams were co-flowed to the droplet-making junction. Droplets 30 μm in diameter were generated by directing the three streams (two aqueous and one oil stream) through a 10 μm nozzle in order to produce droplets at a rate of 2 kHz with one encapsulated cell per three droplets. The library cell and the substrate stream flow rate was 15 μL/h and the oil stream flow rate was 30 μL/h. After emulsification, the droplets were passed through an incubation line (20 min) and reinjected into the sorting device, where more oil was added to the droplet stream at a T-junction (flow rate = 400 µL/h) to increase the inter-droplet spacing. The positive droplets were sorted at an asymmetric Y-junction by measuring the fluorescence and sending several cycles of 20 kHz single-ended square waves to the sorting electrodes using LabView software as previously described [19]. The droplets were sorted onto YNB-CAA Glu agar plates and the cells were recovered by incubation at 30 °C for 48 h. The process was repeated several times to sort the entire library.

### 4.5. ABTS Assay in Agar Plates

Cells recovered on YNB-CAA Glu plates after droplet sorting were incubated at 27 °C for 72 h, replica plated onto YNB-CAA Gal/Raf medium and incubated for an additional 24 h. The screening medium was prepared by mixing a reaction solution containing 333 mM glucose, 1.75 U/mL HRP and 7 mM ABTS with an equal volume of 2% agar. This medium was poured over the cells on the YNB-CAA Gal/Raf plates. Green halos were observed around colonies showing GOx activity.

### 4.6. ABTS Assay in Microtiter Plates

Cells recovered on YNB-CAA Glu plates after droplet sorting were inoculated individually into the wells of 96-well microtiter plates and were cultivated as previously described [37]. We then transferred 5 µL aliquots to new microtiter plates for the ABTS assay, as previously described [25,38,39] with some modifications. The cell aliquots were resuspended in 70 µL phosphate-buffered saline and the OD_600_ was measured. We then added 70 µL 4 mM ABTS containing 1 U/mL HRP. Activity was measured by spectrophotometry at 405 nm every 20 s for 10 min. Two measurements were taken from each culture, both using 4 mM ABTS but containing either 250 or 5 mM glucose. For assay standardization, three wild-type clones were included in each plate. Clones with less than 10% of the wild-type enzyme activity were classed as inactive. The slope of the linear region was calculated for each reaction and normalized to the OD_600_. The mutants showing the highest activity were selected and re-screened.

### 4.7. DNA Isolation and Recloning in P. Pastoris

Plasmid DNA was purified from the best-performing *S. cerevisiae* mutants as previously described [40], the GOx genes were sequenced and then transferred to vector pICZalpha A (Invitrogen) using XhoI/XbaI (New England Biolabs, Ipswich, MA, USA). Competent *P. pastoris* KM71H cells (Invitrogen) were prepared and transformed as previously described [41], and the best-producing clone representing each enzyme variant was selected.

### 4.8. Protein Purification

The transformed *P. pastoris* KM71H cells were cultivated by fermentation for 96 h following the supplier’s recommendations. The cells were then pelleted by centrifugation (11,000× *g*, 10 min, room temperature) using an Avanti J26 XP device (Beckman Coulter, Krefeld, Germany). The supernatant containing GOx was passed through a 0.22 µm PTFE filter (Carl Roth, Karlsruhe, Germany) and concentrated using a Viva Flow 50 system with a 10-kDa membrane (Sartorius, Goettingen, Germany) to a final volume of 5–10 mL. The concentrated supernatant was than dialyzed against 10 mM phosphate buffer pH 6.0 overnight at 4 °C, and loaded onto a 2 mL Fast Flow DEAE Sepharose column (GE Healthcare Europe, Munich, Germany) using an ÄKTA purifier (GE Healthcare, Arlington Heights, IL, USA). The protein was eluted in a linear gradient of 10–250 mM phosphate buffer (pH 6) over 30 column volumes. Collected 50 mL fractions were tested using the ABTS assay and those with separate peaks of GOx activity were collected and concentrated to 5 mL using 10-kDa ultrafiltration columns (Sartorius).

### 4.9. Kinetic Analysis

The kinetic parameters of each GOx variant were determined using triplicate ABTS assays in microtiter plates with different glucose concentrations (1.2–266 mM) at pH 5.5 and 7.4. To determine the K_m_ and k_cat_ values, the slope of each measurement was calculated over the linear region and fitted to a Michaelis-Menten hyperbola using Origin v8 (OriginLab, Northampton, MA, USA). Lineweaver-Burk, Eadie-Hofstee and Hanes-Woolf plots were also constructed, and the outliers were identified and removed. Protein concentration was determined by measuring the absorbance at 280 nm (the absorption of 1.5 mg/mL GOx is considered equivalent to 1 AU based on the sequence, as calculated using ProtParam).

## Figures and Tables

**Figure 1 molecules-25-02418-f001:**
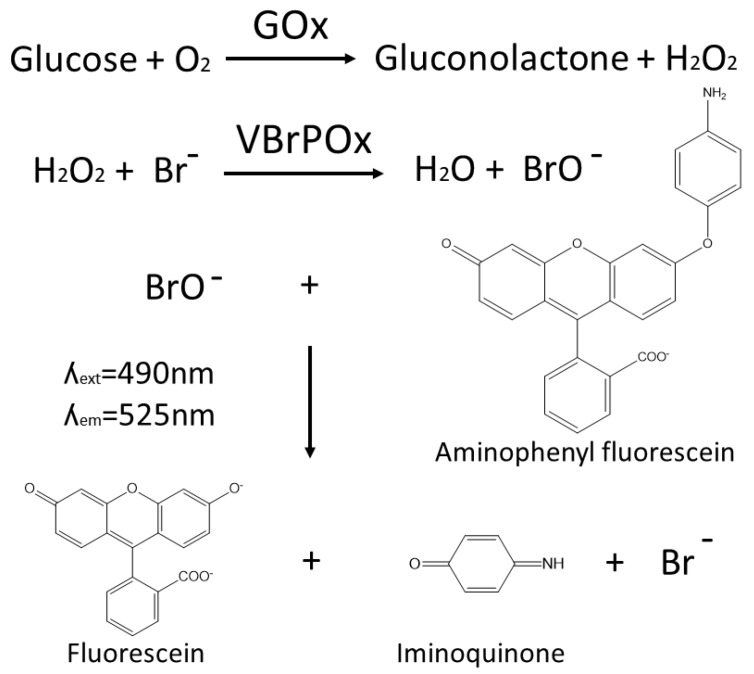
A schematic showing the reaction pathway for the vanadium bromoperoxide-coupled fluorescence assay for the detection of glucose oxidase activity within aqueous microdroplets.

**Figure 2 molecules-25-02418-f002:**
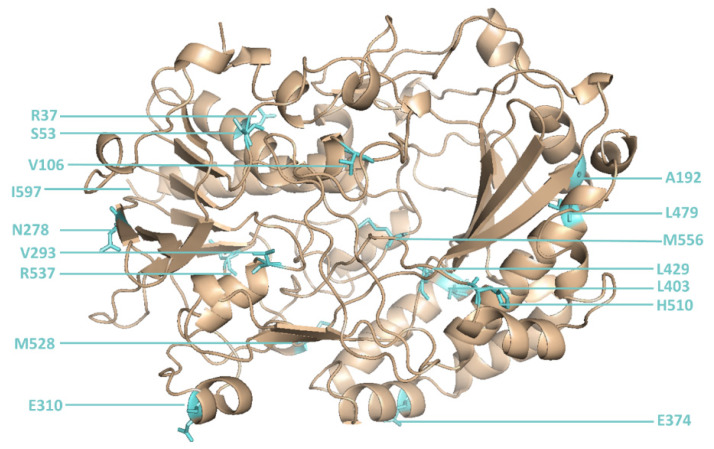
The 16 positions in glucose oxidase that were changed by site-directed mutagenesis.

**Figure 3 molecules-25-02418-f003:**
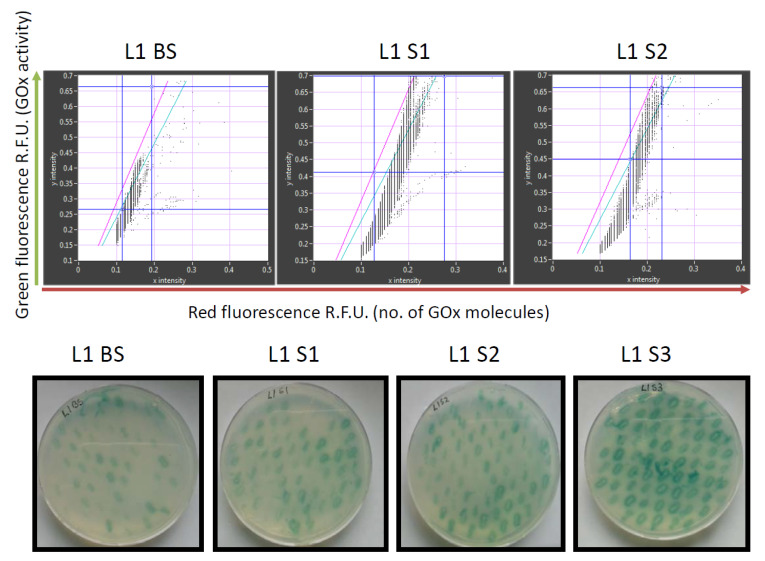
Analysis of glucose oxidase (GOx) library **L1** using the new microfluidics system (upper row) and on standard agar plates (lower row). The library was analyzed before sorting (**BS**), after the first (**S1**), second (**S2**) and third (**S3**) rounds of sorting (**S3**). Red fluorescence (showing the number of GOx molecules) reflects the detection of a C-terminal Myc epitope using a mouse primary anti-c-Myc antibody and a secondary goat anti-mouse IgG antibody labeled with DyLight 550. The sorting gate in the upper panels is between the green and pink lines.

**Figure 4 molecules-25-02418-f004:**
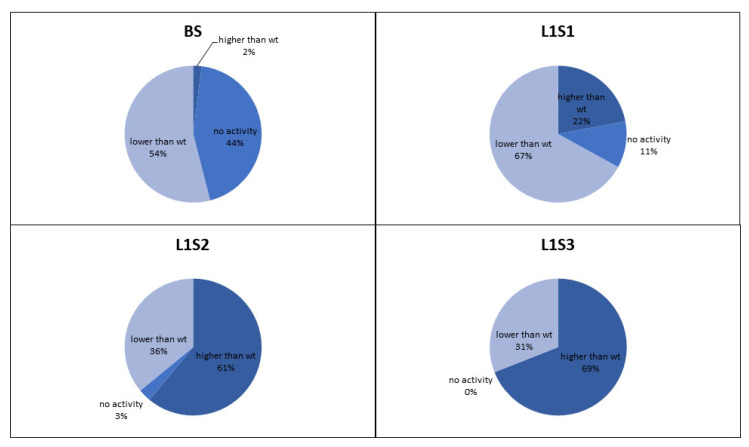
Proportion of glucose oxidase variants in the **L1** library based on activity compared to the wild-type enzyme. The charts show the proportion of variants with no activity, less than wild-type activity, and more than wild-type activity before sorting (**BS**), and after the first (**S1**), second (**S2**) and third (**S3**) rounds of sorting.

**Figure 5 molecules-25-02418-f005:**
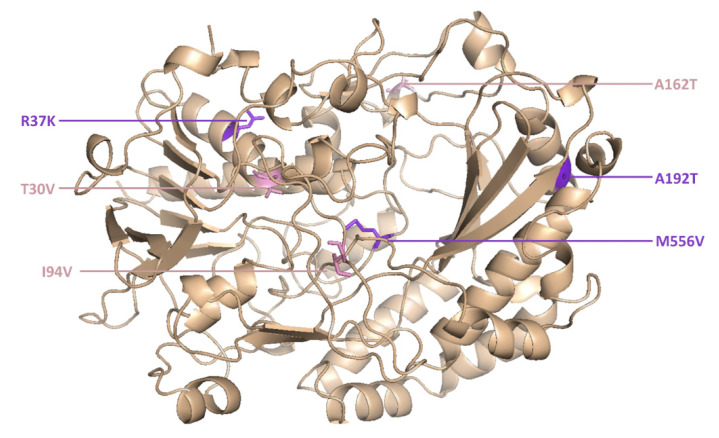
The M6 mutant of glucose oxidase features amino acid substitutions at six positions.

**Table 1 molecules-25-02418-t001:** Kinetic parameters of isolated mutants compared to wild-type glucose oxidase.

Enzyme Variant	K_m_ (mM)	k_cat_ (s^−1^)	k_cat_/K_m_ (mM^−^^1^·s^−^^1^)	S_p_ (U/mg)	Mutations
Wild Type	28.3	189	6.7	86.3	
Parent	15.0	292	19.5	133	T30V, I94V, A162T
A2	18.5	498	26.9	227	T30V, I94V, A162T, R537K, M556V
A21	15.8	466	29.6	212	T30V, I94V, R537K, M556V
M6	18.2	401	22.0	183	T30V, R37K, I94V, A162T, A192T, M556V
F9	19.8	345	17.5	157	T30V, R37K, I94V, V106I, K152R, A162T, M556V
F91	15.1	276	18.3	126	T30V, R37K, I94V, V106I, K152R, A162T
M15	20.2	197	9.75	89.9	T30V, R37K, I94V, A162T, A192T, V293I, I507V, R537K

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
