# Peer review of "A High-Throughput Screening System Based on Droplet Microfluidics for Glucose Oxidase Gene Libraries"

_molecules, 2020, doi:10.3390/molecules25102418_

Round 1

Reviewer 1 Report

The manuscript entitled “A high-throughput screening system based on droplet microfluidics for glucose oxidase gene libraries” by Prodanović et. al describe a new method of high-throughput screening for directed evolution of glucose oxidase (GOx). Authors have successfully developed a microfluidic based method for screening GOx variants for improved activity based on their previously developed assay employing vanadium bromoperoxidase. The manuscript was written well and the study was performed in an elegant manner.

Comments:

  • Page 3, line 103: Authors should mention how did they select the residues for site directed mutagenesis, they should list the selected 16 positions in the supplementary file. It seems there is a typo, as no inactive should be expected if these are site-directed mutagenesis libraries.
  • Authors should also discuss how was the final library created? It is not clear whether the site-saturation library for 16 positions was created by targeting all position at once (saturating all sites simultaneously) or created separately.
  • Page 4, line 117: The reported microfluidics screening method was compared with the previously developed FACS based method. Were the libraries used for sorting the same? How was the quality of the library? Was it analyzed by sequencing random clones before screening? Were all the 16 positions saturated using the QuickChange method?
  • What was the library size? (total number of expected transformants in the library)
  • Authors should discuss the rationale behind site selection for mutagenesis and a figure showing a 3D structure with the sites targeted and the new identified substitutions will help the readers.
  • Figure 2 legend should include a description for the x-axis of the “upper row”. Where is the red fluorescence coming from and how does it corelate to the number of GOx molecules.
  • Page 6, line 165: Authors have claimed up to 97% enrichment of the active clone population using the current method for the site-saturation libraries (where the inactive population is already low when compared to epPCR). Did author check any other GOx library to understand how well the enrichment protocol reproduce?

Author Response

RESPONSE TO REVIEWERS

Reviewer 1

The manuscript entitled “A high-throughput screening system based on droplet microfluidics for glucose oxidase gene libraries” by Prodanović et. al describe a new method of high-throughput screening for directed evolution of glucose oxidase (GOx). Authors have successfully developed a microfluidic based method for screening GOx variants for improved activity based on their previously developed assay employing vanadium bromoperoxidase. The manuscript was written well and the study was performed in an elegant manner.

We thank Reviewer 1 for nice comments about the article and suggestions that improved it.

Comments:

  • Page 3, line 103: Authors should mention how did they select the residues for site directed mutagenesis, they should list the selected 16 positions in the supplementary file. It seems there is a typo, as no inactive should be expected if these are site-directed mutagenesis libraries.

The positions were listed in the supplementary file (Table S1) and we added reference to Table S1 in the main text (Line 114). We observed less active and not active variants.

  • Authors should also discuss how was the final library created? It is not clear whether the site-saturation library for 16 positions was created by targeting all position at once (saturating all sites simultaneously) or created separately.

We added explanation in the main text that the final library was created by multiple-site-directed mutagenesis at all 16 positions simultaneously (Line 115-117).

  • Page 4, line 117: The reported microfluidics screening method was compared with the previously developed FACS based method. Were the libraries used for sorting the same? How was the quality of the library? Was it analyzed by sequencing random clones before screening? Were all the 16 positions saturated using the QuickChange method?

Sorted libraries were the same (Line 134). The library was created using QuickChange by multiple-site-directed mutagenesis method at all 16 positions. Quality was checked by sequencing few random clones before sorting. We added additional explanation for creating library and the quality of it in Materials and Methods section (Lines 225-226, 230-231).

  • What was the library size? (total number of expected transformants in the library)

The library contained 105 different GOx mutants and the number of transformants was 107. We added this info in Results section (Lines 115-117).

  • Authors should discuss the rationale behind site selection for mutagenesis and a figure showing a 3D structure with the sites targeted and the new identified substitutions will help the readers.

Site selection was done using consensus approach and this was stated in Results section (Lines 115-116) with citing the Figure S1 of GOx 3D structure added to supplementary. Figure S2 was also added to Supplementary for M6 mutant.

  • Figure 2 legend should include a description for the x-axis of the “upper row”. Where is the red fluorescence coming from and how does it corelate to the number of GOx molecules.

Red fluorescence is coming from myc epitope at C terminus of GOx that was stained with mouse anti-c-Myc antibodies and with goat anti-mouse DyLight 550 antibodies (Line 239-240). We added explanation in the Figure 2 legend as Reviewer suggested (Lines 122-124).

  • Page 6, line 165: Authors have claimed up to 97% enrichment of the active clone population using the current method for the site-saturation libraries (where the inactive population is already low when compared to epPCR). Did author check any other GOx library to understand how well the enrichment protocol reproduce?

Yes, we sorted reference GOx library made by mixing cells expressing active GOx and cells not expressing GOx in order to determine the optimal laser position during initial round of experiments. We added additional explanation for this part of experiments in Results section (Line 110).

Reviewer 2 Report

In this paper, the authors have identified ans characterized glucose oxidase which is an important industrial enzyme for many applications. 

The results are convincing, and the methods are described adequately.

Although the micro fluidic device has not been described in this paper, readers can easily find it from the references.

This reviewer recommend this paper published as it is.

Author Response

RESPONSE TO REVIEWERS

Reviewer 2

In this paper, the authors have identified ans characterized glucose oxidase which is an important industrial enzyme for many applications. 

The results are convincing, and the methods are described adequately.

Although the micro fluidic device has not been described in this paper, readers can easily find it from the references.

This reviewer recommends this paper published as it is.

We thank Reviewer 2 for his opinion about the article.

Reviewer 3 Report

The mission of the manuscript is to report a high-throughput screening approach based on a microfluidic lab-on-chip device with Yeast surface display for sorting GOx mutants from a saturation mutagenesis library. It’s valuable work. Before the paper could be considered to be accepted for published in Molecules, the authors should make revisions as follow:

  1. Quality of Figure1 and Table 1 should be refined.
  2. In abstract (Line 29-30): the most promising of which (M6) contained a combination of six point mutations that 30 increased the catalytic constant by 2.1-fold compared to wild-type GOx and by 1.4-fold compared to a parental GOx variant. What is the reason? Lower Km, higher Kcat or Kcat/Km?How about A2 and A21 mutants?

Author Response

RESPONSE TO REVIEWERS

Reviewer 3

The mission of the manuscript is to report a high-throughput screening approach based on a microfluidic lab-on-chip device with Yeast surface display for sorting GOx mutants from a saturation mutagenesis library. It’s valuable work. Before the paper could be considered to be accepted for published in Molecules, the authors should make revisions as follow:

  1. Quality of Figure1 and Table 1 should be refined.

Quality of Figure 1, 3 and Table 1 was refined.

  1. In abstract (Line 29-30): the most promising of which (M6) contained a combination of six point mutations that 30 increased the catalytic constant by 2.1-fold compared to wild-type GOx and by 1.4-fold compared to a parental GOx variant. What is the reason? Lower Km, higher Kcat or Kcat/KmHow about A2 and A21 mutants?

The reason was increased kcat. We added explanation in abstract (Line 30). A2 and A21 mutants were previously described (Ref 6).

Reviewer 4 Report

In this paper the authors have used droplet microfluidics for sorting GOx mutants based on enzyme reactions and their resulting fluorescence signals. The authors also report the identification of a novel mutant with increased catalytic constant in comparison to the parental variant as well as the wild-type. However, the paper suffers from serious flaws in terms of clarity of presentation of the results and discussion. Some of the major concerns are as follows-

1) While the introduction defines the significance of droplet microfluidics over existing approaches especially flow cytometry, the authors fail to showcase the significance of GOx and why such a study is necessary. A more detailed explanation of the importance of the GOx enzyme should be mentioned.

Introduction should also include an elaborate description of droplet microfluidics and its different applications.

2) Results lack clarity. While the authors have mentioned the different results there is a lack of quantitative evidences to back up these claims. There is hardly any details of the droplet generator or the sorter  and the mechanisms of their operations has been vaguely mentioned in the results or in the methods.

Insufficient information is provided on the sorting threshold. There is no data provided on the throughput either; number of droplets generated, number of droplets encapsulating the relevant cells, etc.

The authors should resort to more graphical presentation to validate their results.

Also, there is no mention on the experimental reproducibility and it appears that experiments were performed only once.

It is also unclear how the authors identified the 6 point mutation in the M6 variant.

3) While the authors claim the novelty of their microfluidic platform, there is no evidence associated to the claim. The platform seems to be similar to what has been used in the previous studies from the group.

4) Methods section should be more descriptive. While citing prior work is acceptable, it would be good if the authors have a brief description of the platform.

5) What is the throughput of the platform? What volume of samples can be processed in what time?

Minor issues-

1) Typographical errors. Page 6 (line 163)- it could be 'Furthermore in our previous work..'

Author Response

RESPONSE TO REVIEWERS

Reviewer 4

In this paper the authors have used droplet microfluidics for sorting GOx mutants based on enzyme reactions and their resulting fluorescence signals. The authors also report the identification of a novel mutant with increased catalytic constant in comparison to the parental variant as well as the wild-type. However, the paper suffers from serious flaws in terms of clarity of presentation of the results and discussion. Some of the major concerns are as follows-

1) While the introduction defines the significance of droplet microfluidics over existing approaches especially flow cytometry, the authors fail to showcase the significance of GOx and why such a study is necessary. A more detailed explanation of the importance of the GOx enzyme should be mentioned.

Significance of GOx in the food, pharmaceutical, and textile industry and in biofuel cells and biosensors construction was stated in Introduction section with cited references 12-19 that after revision are now references 18-23 (Lines 76-81). In order to additionally emphasize importance of GOx enzyme we added additional sentence citing appropriate references concerning improving GOx properties by protein engineering and high-throughput screening (Lines 81-83)

Introduction should also include an elaborate description of droplet microfluidics and its different applications.

We added paragraph that includes an elaborate description of droplet microfluidics with new cited references 9-16 (Lines 62-68).

2) Results lack clarity. While the authors have mentioned the different results there is a lack of quantitative evidences to back up these claims. There is hardly any details of the droplet generator or the sorter  and the mechanisms of their operations has been vaguely mentioned in the results or in the methods.

More details about droplet generator and the sorter was added into Materials and methods section (Lines 258-260 and 263-266).

Insufficient information is provided on the sorting threshold. There is no data provided on the throughput either; number of droplets generated, number of droplets encapsulating the relevant cells, etc.

Sorting threshold was shown on Figure 2 and information for this was added to the Legend of Figure 2 (Line 124-125). The throughput of 2kHz and number of three droplets per one encapsulated cell were added to Materials and methods (Line 260).

The authors should resort to more graphical presentation to validate their results.

Graphical representation of encapsulation and sorting device was added to Supplementary materials.

Also, there is no mention on the experimental reproducibility and it appears that experiments were performed only once.

Experiment was repeated several times and that info was added to Materials and methods section (Lines 267-268).

It is also unclear how the authors identified the 6 point mutation in the M6 variant.

Additional information was added to Materials and methods section (Line 293).

3) While the authors claim the novelty of their microfluidic platform, there is no evidence associated to the claim. The platform seems to be similar to what has been used in the previous studies from the group.

We claimed that this is a new platform for GOx sorting (Line 22 and 31) and not new microfluidic platform in general. It is based on modification of original GOx ViPer assay adapted for microfluidic setup.

4) Methods section should be more descriptive. While citing prior work is acceptable, it would be good if the authors have a brief description of the platform.

Brief description of the platform was added as Reviewer 4 suggested (Lines 243-247).

5) What is the throughput of the platform? What volume of samples can be processed in what time?

 Throughput of the platform was added to Material and methods section (Line 260).

Minor issues-

1) Typographical errors. Page 6 (line 163)- it could be 'Furthermore in our previous work..'

Typographical error was corrected as Reviewer 4 suggested (Line 180).

Round 2

Reviewer 4 Report

The authors have improved the overall draft nevertheless several grammatical errors can be found throughout the manuscript. While the authors have supplemented some of the concerns raised in the first round of comments, they have not clarified all of the comments well.

1) While the authors have elaborated on the use of microfluidics and their different applications in lines 62-68, these lines are vague and do not specifically respond to the suggestion of elaborating about droplet microfluidics and its different applications.

2) The authors can improve the overall presentation of the results through the use of plots and figures.

Currently there are only 3 figures of which Figure 1 is a reaction which need not have dedicated figure. Instead the authors can incorporate Figure 1 in the text of the method section and could include more relevant figures, data representing plots.Figure 3 is the only quantitative plot in the entire manuscript. The authors can add more quantitative plots for the benefit of the readers

Figure 2, top panel - Analysis of Glucose oxidase library (L1) is blur and the axis legends are not legible. Such figures do not meet the standards for publication in peer reviewed journals such as Molecules.

3) The response of the authors that the platform has not been claimed as a novel microfluidic device and is a new microfluidic platform for GOx sorting (in response to question 3 of the first round of questions) is incorrect.

In the conclusion (line 211), the authors claim that this is a novel microfluidic platform and this needs to be corrected.

4) Moving figures from SI to the main manuscript may be advantageous for the readers and will help improve the overall quality of the paper.

5) Plenty of grammatical and spelling errors can be found throughout the manuscript.

Author Response

1) While the authors have elaborated on the use of microfluidics and their different applications in lines 62-68, these lines are vague and do not specifically respond to the suggestion of elaborating about droplet microfluidics and its different applications.

During the first revision, Reviewer 4 requested that we extend the introduction to cover the importance of GOx and different applications of microfluidics.  We did so by introducing several sentences and references for each subject (marked in red, Lines 62-69 and 93-95).

In order to address the additional requests from Reviewer 4, we have added further information about the advantages of droplet microfluidics over flow cytometry, focusing on aspects that are relevant to the research topic (marked in green Lines 70-85). To avoid overlapping points and redundancy, we have modified the flow of the introduction accordingly. Given that the introduction is already one and a half pages in length and there is further critical analysis of our microfluidics approach in the Discussion section, we believe that this should be enough for research article and any further significant increase would be more appropriate for a review article on microfluidics.

2) The authors can improve the overall presentation of the results through the use of plots and figures.

Currently there are only 3 figures of which Figure 1 is a reaction which need not have dedicated figure. Instead the authors can incorporate Figure 1 in the text of the method section and could include more relevant figures, data representing plots. Figure 3 is the only quantitative plot in the entire manuscript. The authors can add more quantitative plots for the benefit of the readers

We believe that the position of Figure 1 in the results section is important because it shows the adaptation of the ViPer enzymatic assay, which is part of the work we did to establish the new system. We have applied a similar principle in our previous articles on this subject. We believe that Figure 2 is also a quantitative plot because it shows sorting gates on a microfluidic LOC device and specific enzyme activity as a ratio of enzyme activity versus amount  (upper part) and it also shows the number of positive (green) and negative colonies in the library before and after sorting (lower part).

Figure 2, top panel - Analysis of Glucose oxidase library (L1) is blur and the axis legends are not legible. Such figures do not meet the standards for publication in peer reviewed journals such as Molecules.

As requested, we have fixed the blur and legends of Figure 2 and hope that figure now meets the standard for publication in Molecules.

3) The response of the authors that the platform has not been claimed as a novel microfluidic device and is a new microfluidic platform for GOx sorting (in response to question 3 of the first round of questions) is incorrect.

In the conclusion (line 211), the authors claim that this is a novel microfluidic platform and this needs to be corrected.

We corrected statement in Line 211 (now Line 232 marked in green) so that it now states that the device is a new microfluidic platform for GOx sorting.

4) Moving figures from SI to the main manuscript may be advantageous for the readers and will help improve the overall quality of the paper.

We moved the figures from SI to the main manuscript as suggested to improve the overall quality of the paper.

5) Plenty of grammatical and spelling errors can be found throughout the manuscript.

The revised manuscript has been comprehensively edited by a professional scientific writer and native English speaker (Dr. Richard M Twyman, TRM Ltd).